# Engineered Adoptive T-Cell Therapies for Breast Cancer: Current Progress, Challenges, and Potential

**DOI:** 10.3390/cancers16010124

**Published:** 2023-12-26

**Authors:** Diego F. Chamorro, Lauren K. Somes, Valentina Hoyos

**Affiliations:** 1Center for Cell and Gene Therapy, Baylor College of Medicine, Houston, TX 77030, USA; diego.chamorro@bcm.edu (D.F.C.); lauren.somes@bcm.edu (L.K.S.); 2Dan L. Duncan Comprehensive Cancer Center, Baylor College of Medicine, Houston, TX 77030, USA

**Keywords:** breast cancer, immunotherapy, CAR T, TCR T, clinical trials

## Abstract

**Simple Summary:**

Breast cancer stands as the predominant form of cancer identified in women, underscoring the urgent demand for innovative targeted therapies. Engineered adoptive cell therapies represent a groundbreaking approach, empowering the immune cells of patients to precisely target their tumors through cancer-specific receptors. Encouragingly, preclinical studies have illuminated the immense promise of this strategy. Nonetheless, the translation of this technique into clinical practice hinges on the accumulation of additional robust clinical data. Within this review, we offer a comprehensive examination of the current landscape surrounding engineered cell therapies for breast cancer, delving into both their limitations and the compelling prospects for enhancement.

**Abstract:**

Breast cancer remains a significant health challenge, and novel treatment approaches are critically needed. This review presents an in-depth analysis of engineered adoptive T-cell therapies (E-ACTs), an innovative frontier in cancer immunotherapy, focusing on their application in breast cancer. We explore the evolving landscape of chimeric antigen receptor (CAR) and T-cell receptor (TCR) T-cell therapies, highlighting their potential and challenges in targeting breast cancer. The review addresses key obstacles such as target antigen selection, the complex breast cancer tumor microenvironment, and the persistence of engineered T-cells. We discuss the advances in overcoming these barriers, including strategies to enhance T-cell efficacy. Finally, our comprehensive analysis of the current clinical trials in this area provides insights into the future possibilities and directions of E-ACTs in breast cancer treatment.

## 1. Introduction

Breast cancer is the most common cancer diagnosed in women and the second most common cause of cancer-related death [1]. Clinically, it is divided into four molecular subtypes based on expression of estrogen receptor (ER), progesterone receptor (PR), and human epidermal growth factor receptor 2 (HER2) [2]. Various therapeutic modalities that target these receptors are utilized in the treatment of breast cancer, including anti-HER2 monoclonal antibodies and selective ER degraders (SERDs) [3,4]. Despite the great success of these therapies, there is a need for novel targeted treatments that can benefit a wider variety of breast cancer patients.

Engineered adoptive T-cell therapies (E-ACTs), a subset of immunotherapy, equip a patient’s T-cells with engineered receptors that specifically recognize their cancer. These therapies have revolutionized the treatment of hematologic malignancies; however, none have successfully emerged as a clinically relevant option for breast cancer. A thorough understanding of the factors that limit the success of engineered adoptive T-cell therapies in breast cancer is vital to improving their therapeutic outcomes. In this review, we analyze the main types of E-ACTs for breast cancer, their limitations and challenges, and the current landscape of these therapies in clinical trials.

## 2. Engineered Adoptive T-Cell Therapies for Breast Cancer

E-ACTs are a type of immunotherapy in which the patient’s immune cells are modified to confer a customized immune response to their cancer [5,6,7]. These therapies can be divided into two major categories: chimeric antigen receptor (CAR) T-cell therapy and T-cell receptor (TCR) T-cell therapy. Both approaches are utilized in breast cancer models with varying degrees of success. Table 1 compares TCR T and CAR T-cell receptors regarding their MHC restriction, sensitivity, antigens recognized, and co-stimulatory molecules.

CAR T-cell therapies combine the specificity of a monoclonal antibody (mAb) with the signaling components of a TCR, resulting in a synthetic receptor that is not major histocompatibility complex (MHC)-restricted (Figure 1) [11]. The typical CAR is composed of a single-chain variable fragment (scFv) fused to hinge and transmembrane domains, followed by an intracellular signaling domain [12]. The scFv is derived from the heavy and light chain variable regions of a mAb and is responsible for antigen recognition [13]. The intracellular signaling domain of first-generation CARs is comprised solely of the CD3ζ or FcRɣ signaling domains; clinical efficacy with these first-generation CARs was limited, however, as it was proven that they do not produce a durable anti-tumor response due to a lack of expansion and persistence [14,15]. Second- and third-generation CAR constructs incorporate the signaling domains of known T-cell co-stimulatory molecules. The two most common co-stimulatory domains used are CD28 and 4-1BB (CD137). Others include inducible T-cell co-stimulator (ICOS), CD27, MyD88, CD40, and OX40 (CD134) [16,17]. Fourth-generation CARs, also known as T-cells redirected for universal cytokine killing (TRUCKs), are engineered to incorporate cytokines or their receptors, which serve to support T-cell activity and survival [18,19], resulting in more durable T-cell responses [20].

TCR T-cell therapies, on the other hand, utilize naturally occurring TCRs isolated from T-cell clones that recognize the tumor antigen of interest. The α and β chains of the isolated TCR are expressed in the recipient’s T-cells, which dimerize and associate with endogenous CD3ε/ɣ/δ/ζ subunits to confer the desired specificity (Figure 1) [21]. Engineered TCR (E-TCR) T-cells can recognize peptides from both intracellular and extracellular tumor antigens presented on surface MHC molecules, including neoantigens arising from tumor-specific mutations [6,21]. Since TCRs recognize tumor antigens in an MHC-dependent manner, E-TCRs are matched to the patient’s expressed MHC alleles [22]. As a significant portion of breast cancer cases have mutations in *PIK3CA*, *TP53*, and *ESR1* [21,23,24], the use of E-TCRs specific for these mutations is an attractive therapeutic avenue.

Compared to the success seen in hematologic malignancies, E-ACTs for breast cancer are limited in their efficacy and feasibility. Six CAR T-cell products have been approved by the US Food and Drug Administration (FDA) for B-cell malignancies and multiple myeloma [25]; currently, there are no FDA-approved CAR T-cell therapies for solid tumors. In the case of TCR T-cells, there are no FDA-approved TCR T-cell products for hematologic or solid tumors to date, and there is little clinical data regarding their efficacy in breast cancer [21]. Many factors limit the clinical success of engineered adoptive cell therapies in breast cancer, including difficulties identifying suitable tumor targets, the immunosuppressive tumor microenvironment, diminishing T-cell persistence, and the costs associated with treatment.

## 3. Identification of Suitable Tumor Targets

The ideal target antigen for engineered cell therapies is overexpressed by the tumor and absent from normal cells in surrounding tissue and vital organs. A wide variety of breast cancer targets have been identified; however, many of these targets are also expressed on normal cells to a certain degree. Therefore, careful antigen selection is a crucial aspect of E-ACTs to ensure treatment efficacy and patient safety. The three main classes of tumor antigens are tumor-associated antigens (TAAs), cancer-germline antigens (CGAs), and tumor-specific antigens (TSAs) [6,26,27].

TAAs are expressed in malignant cells and some healthy tissues. As a result, targeting these antigens carries the risk of on-target/off-tumor toxicities. This risk, however, can be managed with proper TAA selection. TAAs can be classified as either differentiation or overexpressed antigens. Differentiation antigens are expressed in the tumor and the corresponding healthy tissue and are thus associated with the most significant risk of on-target/off-tumor toxicities [21,28]. Most breast cancer TAAs are overexpressed antigens. Overexpressed antigens are enriched in tumor cells, with minimal expression in healthy cells; however, there is still a risk of on-target/off-tumor toxicity. To overcome this risk, engineered cell therapies can be optimized to necessitate a high antigen density for receptor activation, therefore minimizing the destruction of normal cells with low antigen density [29].

CGAs, also referred to as cancer/testis antigens (CTAs), are a unique group of TAAs that are expressed exclusively in germ cells and absent from normal somatic tissues [30,31]. Their expression in malignant cells results from aberrant gene expression due to DNA hypomethylation [30,32]. CGAs are desirable targets for E-ACT due to their restricted expression pattern. Potential CGA targets for breast cancer have been evaluated for CAR T and TCR T-cell therapies.

Finally, TSAs, as the name suggests, are exclusively expressed in malignant cells. These antigens may arise from mutations (neoantigens) or the expression of viral elements [28,33]. Neoantigens are ideal targets for TCR T-cell therapy as TCRs can detect mutant peptides harboring a single-point mutation without reactivity to the wildtype peptide [34]. While many mutations are patient-specific or “private” neoantigens, driver mutations in certain genes can result in “public” neoantigens that are shared among many breast cancer patients [35].

Table 2 summarizes the most common breast cancer TAA, CGA, and TSA targets under investigation.

### Remaining Challenges: Antigen Heterogeneity and On-Target/Off-Tumor Toxicity

In solid tumors, including breast cancer, many target antigens are not uniformly expressed among all cells within the tumor. As a result, patients may initially respond to therapy but later progress due to the outgrowth of antigen-negative tumor cells [118]. HER2 intratumoral heterogeneity, for example, is reported in up to 40% of breast cancers and is a potential mechanism for resistance [119]. Strategies to overcome intratumoral heterogeneity in solid and hematologic malignancies include using epigenetic modulators to increase surface expression of target antigens [120,121,122] and targeting multiple antigens expressed throughout the tumor simultaneously [120,123,124,125]. Few preclinical studies using these approaches to address tumor heterogeneity in breast cancer models exist, and they will be necessary for the success of these therapies in breast cancer. An early study in a murine breast cancer model found that expression of a murine CGA could be increased by treating the cells with an epigenetic drug that inhibits DNA methylation [126]. More studies are needed to test if these treatments may be combined with E-ACTs in in vivo models.

Multi-antigen targeting also reduces the risk of on-target/off-tumor toxicities. While few preclinical studies utilize breast cancer models, a recent study presented an elegant method for restricting CAR T-cells to dual antigen encounters using targets commonly expressed in breast cancer. These logic-gated intracellular network (LINK) CARs only mount an anti-tumor response when both target antigens are present, thus significantly limiting the potential for on-target/off-tumor toxicity [127]. Future studies are needed, however, to test the efficacy of this approach in solid tumor models.

## 4. Overcoming the Tumor Microenvironment

The breast cancer tumor microenvironment (TME) consists of various suppressive immune cells, stromal cells, and soluble components that together play a vital role in the growth, survival, and spread of malignant cells (Figure 2) [128,129]. The TME is one of the most significant barriers for E-ACTs to overcome, as T-cells must not only perform their anti-tumor functions but also navigate the hostile TME milieu. Efforts to further dissect the complexity of the TME via single-cell RNA sequencing (scRNAseq) analysis have identified various subclusters of immune and stromal cells that perform a variety of pro-tumoral functions [130,131,132,133]. These studies also identify a wealth of potential prognostic markers and therapeutic targets [132,133,134,135,136]. As the full complexity of the TME is beyond the scope of this review, we will summarize the primary TME components that contribute to the inhibition of E-ACTs.

### 4.1. Immune Microenvironment of Breast Cancer

The breast cancer TME is divided into different compartments based on their cell composition and proximity to the tumor cells [128,129]. The local, or intratumoral, compartment primarily includes several types of immune cells, including regulatory T-cells (T_regs_), myeloid-derived suppressor cells (MDSCs), and tumor-associated macrophages (TAMs). These cell types perform specific functions that promote tumor growth and suppress anti-tumor immunity.

#### 4.1.1. Regulatory T-Cells (T_regs_)

T_regs_ represent one of the primary immune populations that favor immunotolerance in the breast TME. T_regs_ are characterized by their expression of the transcription factor Foxp3 and are recruited to the TME via chemokines secreted by the breast tumor cells [128,137,138]. T_regs_ suppress the functions of effector cells through several mechanisms, including the secretion of IL-10, TGFβ, and adenosine, competitive consumption of IL-2, and expression of CTLA-4/PD-L1 [139,140,141]. While the prognostic significance of T_regs_ in breast cancer reportedly varies among molecular subtypes [139], studies have identified relationships between T_reg_ infiltration and certain prognostic variables. An analysis of tumor samples from various molecular breast cancer subtypes found that elevated numbers of T_regs_ are associated with aggressive tumor phenotypes, larger tumor size, and estrogen receptor negativity [142,143]. Similarly, a study of breast tumor resident T_regs_ found that their frequency increases in TNBC. These T_regs_ also express high levels of chemokine receptor 8 (CCR8), which is associated with higher-grade tumors and poor survival [144].

Due to their prevalence and immunosuppressive capabilities, T_regs_ are a desirable target for improving the efficacy of E-ACTs. Broadly, methods to inhibit the effects of T_regs_ include immune checkpoint inhibitors against various co-inhibitory molecules (CTLA-4, PD-1, LAG-3, TIM-3, and TIGIT), depletion via targeting of T_reg_-specific surface molecules, agonistic antibodies against tumor necrosis factor receptor superfamily molecules (GITR, 4-1BB, OX40, and others), and small molecule drugs targeting characteristic features of T_regs_ [145,146]. In breast cancer, genetic depletion of T_regs_ significantly enhanced the efficacy of checkpoint inhibition in a claudin-low TNBC model; however, the use of pharmacologic methods to deplete T_regs_ also reduced the numbers of infiltrating CD4^+^ and CD8^+^ T-cells [147]. The histone deacetylase (HDAC) inhibitor vorinostat was found to decrease the number of Foxp3^+^ cells in syngeneic 4T1 TNBC tumors and potentiate the effect of checkpoint inhibition [148]. Studies utilizing cyclophosphamide and letrozole found that both drugs can reduce T_reg_ populations in peripheral blood and in the tumor [149,150]. Additionally, a phase I study of daclizumab, an anti-CD25 mAb, in combination with peptide vaccination reported significant depletion of CD25^+^Foxp3^+^ T_regs_ and CD8 responses to the tumor peptides, despite CD25 expression on effector T-cells [151]. Currently, no preclinical studies focus on targeting T_regs_ in the context of E-ACTs, likely due to difficulties specifically targeting the T_regs_ while sparing the engineered T-cells.

#### 4.1.2. Myeloid-Derived Suppressor Cells (MDSCs)

MDSCs are a heterogeneous group of immature myeloid cells that arise due to altered myelopoiesis driven by chronic inflammation in cancer and primarily function to suppress T-cell-mediated immune responses [152,153]. They are divided into two main phenotypic subtypes: monocytic MDSCs (M-MDSCs) and granulocytic/polymorphonuclear (G-/PMN-MDSCs). While phenotypic characterizations can vary, M-MDSCs are typically defined as CD11b^+^CD14^+^HLA-DR^−/low^CD15^−^ cells, and PMN-MDSCs can be defined as CD11b^+^CD14^−^CD15^+^ or CD66b^+^CD15^+^CD14^−/dim^CD33^dim^HLA-DR^−^ cells [152,154,155]. Both subtypes of MDSCs maintain an immunosuppressive environment through a variety of mechanisms, including deprivation of metabolic fuels required by T-cells, induction of oxidative stress, recruitment of T_regs_, and expression of high levels of PD-L1, to name a few [153,156]. M-MDSCs predominantly mediate their immunosuppressive effects through elevated ARG1, iNOS, and TGFβ expression. On the other hand, PMN-MDSCs dysregulate T-cell function through cell-to-cell contact and ROS production [153].

In metastatic breast cancer, high levels of M-MDSCs are associated with aggressive disease and shorter survival [157,158]. M-MDSCs have been shown to suppress CAR T-cell efficacy in vitro and orthotopic mouse models [52,159]. Preclinical studies describe various methods to target MDSCs in breast cancer models. Drugs that favor the differentiation of MDSCs into mature, less suppressive cell types, such as all-trans retinoic acid, can diminish the immunosuppressive microenvironment [160]. In addition, the HDAC inhibitor entinostat attenuates the immunosuppressive function of G-/PMN-MDSCs in combination with immune checkpoint inhibitors in transgenic breast cancer models [161]. In the context of CAR T-cell therapy, polyinosinic-polycytidylic acid (poly I:C), a ligand of TLR3, not only enhances CAR T-cell function but also decreases MDSC levels in peripheral blood and attenuates their immunosuppressive activity [159]. Furthermore, olaparib suppresses MDSC recruitment through the SDF1α/CXCR4 axis and enhances CAR T-cell efficacy in syngeneic breast cancer models [162].

Methods to arm engineered T-cells with other chimeric receptors that provide co-stimulation while targeting immunosuppressive cell types have also been explored. MDSCs express a receptor called tumor necrosis factor (TNF)-related apoptosis-inducing ligand (TRAIL) receptor 2 (TR2). This receptor induces apoptosis upon TRAIL engagement and has previously been targeted using an agonist mAb [163,164]. A novel co-stimulatory receptor composed of the scFv of the TR2 mAb combined with a 4-1BB endodomain successfully protected anti-MUC1 CAR T-cells from MDSC immunosuppression and promoted superior anti-tumor activity in breast cancer models [52].

#### 4.1.3. Tumor-Associated Macrophages (TAMs)

TAMs are another immunosuppressive tumor-associated myeloid cell type. TAMs are classified into two primary phenotypes that depend on cytokine exposure. M1 macrophages, which possess anti-tumor functions, are stimulated by IFNɣ and TNF. M2 macrophages, on the other hand, activated by IL-4, IL-10, and IL-13, promote tumor growth and suppress anti-tumor immune responses [129]. TAMs suppress T-cells by secreting factors such as IL-10, ARG1, iNOS, PGE2, and TGFβ and recruiting T_regs_ [165,166,167]. TAMs can also express PD-L1, CD80/86, or death receptor ligands that engage with inhibitory receptors on effector cells [166].

In breast cancer, high levels of TAMs are associated with metastasis, lower rates of survival, and overall poor prognosis [168,169]. Preclinical studies suggest that TAMs may be depleted from the breast TME via CARs targeting proteins expressed by both the tumor and TAMs. One potential target, AXL, is a receptor tyrosine kinase expressed in both breast cancer cells and TAMs. Anti-AXL CAR T-cells demonstrate significant anti-tumor activity in TNBC models and have the potential to overcome the immunosuppressive microenvironment by inhibiting TAM cytokine secretion [26,97,98,170]. Another receptor tyrosine kinase, ephrin receptor A10 (EphA10), is detected in TNBC cells, TAMs, and MDSCs. EphA10-specific CAR T-cells have been shown to inhibit in vivo tumor growth in an orthotopic MDA-MB-231 tumor model. Although the effects of these CAR T-cells remain to be tested in immunocompetent models, anti-EphA10 antibodies increased the infiltration and activity of cytotoxic T-cells in a syngeneic 4T1 tumor model, suggesting that blocking EphA10 on TAMs/MDSCs restores T-cell activity [171]. In addition, pharmacologic inhibition of sphingosine 1-phosphate receptor 3 (S1PR3), a bioactive lipid molecule expressed in breast cancer, resulted in TME remodeling via the recruitment of pro-inflammatory macrophages and improved the efficacy of anti-EpCAM CAR T-cell therapy in a murine breast cancer model [61]. Other potential therapeutic avenues to target TAMs in the breast TME include depletion via CSF-1/CSF1R inhibition [172,173,174,175], modulation via class IIa HDAC inhibition [176], and reprogramming via antibodies targeting the pattern recognition scavenger receptor MARCO [177]. Further studies, however, are needed to assess whether these methods can bolster the efficacy of breast cancer E-ACTs.

### 4.2. Non-Immune Microenvironment of Breast Cancer

In addition to immune cells that suppress effector functions and support tumor growth, the regional, or breast, compartment of the TME is home to other physical and structural components that influence tumor and T-cell behavior. Acellular components such as the extracellular matrix and hostile metabolic conditions, and cellular components such as cancer-associated fibroblasts (CAFs) and endothelial cells, provide structural support for tumor cells, serve as a barrier for anti-tumor immunity, and provide nutrients, proliferative stimuli, and tumor niche protection.

#### 4.2.1. Extracellular Matrix (ECM)

Breast tumors are encapsulated by a complex and dynamic ECM composed of collagens, fibronectin, laminins, glycosaminoglycans and proteoglycans, matricellular proteins, and ECM remodeling enzymes [178]. Collagen within the breast ECM plays a significant role in cancer progression and metastasis [179,180,181,182] and presents a physical barrier that adoptively transferred T-cells must traverse to infiltrate the tumor tissue [183]. Discoidin domain receptor 1 (DDR1) provides further ECM fortification by promoting collagen fiber alignment. This, in turn, suppresses anti-tumor immunity and promotes breast cancer growth by preventing T-cell infiltration [183,184]. Antibodies to neutralize DDR1 effectively disrupt collagen alignment and support anti-tumor immunity [183]. Furthermore, macrophages are an essential source of matrix metalloproteinases (MPPs) that degrade ECM components. In a preclinical study, macrophages engineered to express an anti-HER2 CAR significantly inhibited HER2-4T1 tumor growth in an immunocompetent model [185].

#### 4.2.2. Cancer-Associated Fibroblasts (CAFs)

CAFs are derived from normal fibroblasts activated by tumor-derived inflammation and are the most prominent cell type in the breast TME. CAFs perform various functions, including remodeling the TME, promoting tumor malignancy and angiogenesis, suppressing immune cells, and acting as a physical barrier to infiltrating T-cells [186,187,188]. Breast cancer CAFs express high levels of stromal cell-derived factor-1 (SDF-1), which is responsible for promoting angiogenesis via recruitment of endothelial progenitor cells (EPCs) and stimulating tumor growth [189]. In addition, CAF-derived exomes contain micro RNAs (miRNAs) that promote breast cancer progression and metastasis [190].

Due to their prevalence in the breast TME and pro-tumoral functions, CAFs are an attractive target for E-ACTs. CAR T-cells targeting fibroblast activation protein (FAP) have been evaluated in syngeneic 4T1 TNBC mouse models with conflicting results. One study found that FAP-specific CAR T-cells demonstrated minimal anti-tumor activity and caused bone toxicities due to expression of FAP on bone marrow stromal cells [191]. Other preclinical studies of anti-FAP CAR T-cells derived from other mAb clones report significant anti-tumor activity in 4T1 [192] and other solid tumor models [193], with no on-target/off-tumor toxicities. Despite limited success in breast cancer thus far, methods for targeting CAFs to improve T-cell efficacy are being actively investigated. A recent study found that CAFs secrete high levels of IL-6, increasing PD-L1 expression in TNBC cells and inhibiting CAR T-cell efficacy. The authors recommend the exploration of methods to target the signaling pathways driving IL-6 and PD-L1 expression to improve the response to CAR T-cell therapy [80].

#### 4.2.3. Endothelial Cells

Endothelial cells within the TME play a pivotal role in tumor angiogenesis through their expression of vascular endothelial growth factor (VEGF) and vascular endothelial growth factor receptors (VEGFR) [194,195]. Tumor-associated endothelial cells also hinder T-cell extravasation by suppressing the required endothelial adhesion molecules [196]. In breast cancer, tumor cells can promote mesenchymal phenotypes in endothelial cells, which favor tumor proliferation, stemness, and invasiveness [197]. Due to their pivotal role in the breast TME, endothelial cells are a prime target for engineered T-cell therapies. CAR T-cells specific for tumor endothelial marker 8 (TEM8), a cell surface protein that functions in endothelial cell migration and invasion, are effective not only against TNBC cells but also against the associated tumor endothelium [198]. Furthermore, a ligand-based CAR utilizing VEGF-C as the antigen binding domain effectively targeted VEGFR-2 and VEGFR-3-positive breast cancer cells and the tubular structures formed by human umbilical vein endothelial cells (HUVECs), thus inhibiting angiogenesis [194]. Moderate anti-tumor effects were seen in a study utilizing VEGFR-2-specific CAR macrophages. However, the authors did not evaluate their impact on the associated endothelial cells [199].

#### 4.2.4. Metabolic Conditions

Hypoxia and competition for metabolic fuels within the TME contribute to exacerbated immunosuppression and poor T-cell survival. In the hypoxic TME, tumor cells, immune cells, and other cellular components constantly compete for limited amounts of metabolic fuels and nutrients [200]. The lack of essential nutrients and low availability of oxygen force T-cells to adopt an anaerobic metabolism, hindering full T-cell activation. In addition, other cellular components of the TME can deplete amino acids essential for T-cell activation and proliferation, such as arginine and cysteine, and release reactive oxygen species that hinder T-cell signaling [201]. Tumor cells can produce high levels of metabolites like adenosine and lactate, leading to T-cell inhibition [202]. CAFs and mesenchymal cells can also produce toxic metabolites for T-cells [203], and it has been shown that selective depletion of those cell populations diminishes the immunosuppressive conditions and improves T-cell metabolic function [204,205,206].

Other strategies to combat the effects of these “chemical” immunosuppressors have been tested in preclinical breast cancer models. For example, adenosine exerts immunosuppressive effects in T-cells through the A2a receptor (A2aR) [207]. mRNA and protein analysis of breast tumor samples also revealed that A2aR expression is associated with aggressive phenotypes, poor survival, and immunosuppressive immune infiltrates [208]. CRISPR/Cas9 knockout of A2aR can increase the effector function of CAR T-cells without compromising their memory phenotype or persistence. A2aR knockout CAR T-cells mediated an enhanced therapeutic response in a HER2+ murine breast cancer model [209].

## 5. Persistence of Adoptively Transferred Engineered Cells

While E-ACTs have the potential to become a powerful component of the breast cancer treatment arsenal, success is limited in part due to a lack of cell persistence at the tumor site. Compared to cell therapies for hematologic malignancies, which encounter the cancer cells immediately upon entering the bloodstream, T-cells redirected against breast cancer must endure long enough to navigate to and penetrate the immunosuppressive TME. Only after this do they encounter their target antigen. Moreover, for successful tumor debulking and elimination, these cells must not only reach the tumor but also thrive and remain within it until the clearance of malignant cells. Several strategies have been utilized to support CAR T-cell persistence and expansion in breast cancer models (Figure 3).

### 5.1. Engineered Chimeric Receptors

One approach to improve T-cell persistence in suppressive breast cancer TME is the inclusion of engineered receptors that provide additional cytokine signaling or co-stimulation. This added support can help CAR T-cells resist inhibitory signals. Inverted cytokine receptors, for instance, provide proliferative signals in response to immunosuppressive cytokines in the TME, turning the tumor’s defenses against it. IL-4 is an inhibitory Th2 cytokine that has been shown to promote the survival of breast cancer cells and support T_regs_ [210,211,212]. An inverted cytokine receptor composed of the IL-4 receptor exodomain fused to the IL-7 receptor endodomain improved the persistence and anti-tumor activity of anti-MUC1 CAR T-cells against an IL-4-secreting breast cancer CDX model [53]. As the immunosuppressive TME typically contains low levels of the immunostimulatory cytokines necessary to maintain CAR T-cell activation, IL-7 signaling has also been provided as an engineered constitutively active receptor known as C7R [213]. C7R activates the IL-7 signaling axis without needing extracellular cytokines and enhances the anti-tumor activity of anti-AXL CAR T-cells in TNBC models [98,213]. Currently, three E-ACT clinical trials include C7R co-stimulation (NCT04099797, NCT03635632, NCT04664179).

### 5.2. Soluble Cytokine Production

CAR T-cells engineered to secrete specific cytokines are also known as TRUCK (T-cells Redirected towards Universal Cytokine Killing) CAR T-cells [187,214]. TRUCK CAR T-cells incorporating various soluble cytokines, including IL-15, IL-7, and IL-18, have been explored for breast cancer.

#### 5.2.1. Interleukin-15 (IL-15)

While IL-15 is structurally similar to IL-2, it possesses certain functions in vivo that distinguish it from IL-2 and make it a desirable candidate for enhancing E-ACT efficacy [215]. Unlike IL-2, IL-15 does not affect T_reg_ expansion. It regulates tumor-infiltrating lymphocyte numbers within the TME and plays a crucial role in T-cell activation, expansion, differentiation, and function [215,216,217]. In a recent preclinical study, IL-15 co-expression enhanced the persistence and anti-tumor activity of anti-EGFRvIII CAR T-cells in a murine breast cancer model [218]. These CAR T-cells also expressed CXCR2, which accelerated T-cell trafficking to the tumor site via chemokines expressed in the breast TME [218]. Additional studies in other solid tumor models also demonstrate that IL-15 co-expression is a powerful method to enhance T-cell proliferation and anti-tumor activity in the solid TME [216,217,219,220,221,222]. These encouraging results will hopefully inspire additional breast cancer studies and lead to clinical investigations.

#### 5.2.2. Interleukin-7 (IL-7)

As with IL-15, IL-7 is a critical player in the expansion of naïve and memory T-cells and does not affect T_regs_ [223,224]. In preclinical breast cancer studies, IL-7 is often combined with chemokines such as CCL19 and CCL21 to improve the chemotaxis of CAR T cells and other immune cells to the tumor site. In combination with CCL21, IL-7 improved the anti-tumor activity of CLDN18.2-specific CAR T-cells in a syngeneic mouse model without lymphodepletion [225]. Another study found that anti-Folate Receptor α (FRα) CAR T-cells co-expressing IL-7 and CCL19 demonstrated superior T-cell infiltration [226]. The addition of CCL19 and CCL21 increased the infiltration of endogenous dendritic cells [225,226], while CCL21 also showed inhibition of tumor angiogenesis [225]. IL-7 is also being investigated in other solid tumor CAR T-cell models [227,228,229].

#### 5.2.3. Interleukin-18 (IL-18)

IL-18 is a pleiotropic cytokine with various functions in different T-cell types. Most notably for CAR T-cell therapies, IL-18 induces IFNɣ production, decreases T_regs_, and increases the expansion of CD8^+^ T-cells [230,231,232]. Ruixin and colleagues’ previously mentioned study of anti-EGFRvIII CAR T-cells with CXCR2 also included conditions with IL-18 co-expression (as opposed to IL-15). Similar to their results for IL-15, anti-EGFRvIII CAR T-cells co-expressing IL-18 had reduced expression of exhaustion markers and superior anti-tumor activity [218]. Additional studies in other solid tumor models have also produced promising results [231,233,234,235].

## 6. Cost of Autologous Therapy

One of the significant challenges of autologous E-ACT is the cost. Autologous E-ACT requires a lengthy and complex manufacturing process that must be carefully orchestrated and tailored to each patient. For example, current FDA-approved CAR T-cell therapies range from $373,000 to $475,000 for a single infusion [236,237,238]. Overall, the exorbitant costs of E-ACT can be improved by lower-cost manufacturing techniques and allogeneic, or “off-the-shelf,” therapies. Such techniques are beginning to be explored in the context of breast cancer.

### 6.1. Non-Viral Manufacturing Techniques

Genetic modifications for cellular immunotherapies are commonly introduced via viral transduction. While this method results in high transduction efficiencies, it is extremely labor intensive, has limited capacity for multigene insertions, and is expensive to produce clinically [239,240,241]. Non-viral manufacturing techniques, such as transposons and in vitro-transcribed (IVT) mRNA, have emerged as potential low-cost alternatives.

#### 6.1.1. Transposon Systems

DNA transposons are mobile genetic elements excised from one part of the genome and integrated into another using a “cut and paste” mechanism mediated by a transposase enzyme [242]. Sleeping Beauty (SB) is one of the most widely used transposon systems for genetically modifying human cells. SB is a synthetic transposon derived from inactive transposon sequences in fish genomes [243,244]. The SB vector system comprises two components: the transposon DNA, consisting of the gene of interest flanked by inverted terminal repeats (ITRs), and the SB transposase [243,244]. The SB transposase recognizes the ITR sequences and transfers the transgene from the donor vector to the acceptor site in the genome [244]. Much of the preclinical validation of SB as a viable method for genetic manipulation in E-ACT is focused on CAR T-cell therapy for hematological malignancies and is well-reviewed by Moretti et al. [244].

Compared to viral vectors, the production of transposon plasmids under GMP conditions is significantly cheaper and faster. Using transposons also does not require the complex biohazard procedures associated with viral vector production [242]. Despite these advantages, few clinical trials evaluating E-ACT for breast cancer utilize transposon systems. As of July 2023, two clinical trials use the SB system for CAR/TCR T-cell therapy for breast cancer (NCT04102436, NCT05694364). These trials are in the recruiting phase, with no results posted.

#### 6.1.2. In Vitro-Transcribed (IVT) mRNA

Another alternative to viral manufacturing techniques is IVT mRNA. While the process mirrors viral transduction, IVT mRNA is unique in that genetic modification is transient, allowing for enhanced safety and the ability to modulate expression levels [240,244,245,246]. After isolation and expansion of patient T-cells, IVT mRNA encoding the construct of interest, such as a CAR, TCR, or cytokine, is electroporated into the cells. After mRNA translation is confirmed, the cells are re-infused into the patient [245]. Early preclinical papers demonstrate that electroporation of TCR-encoding IVT mRNA into T-cells produces cytotoxic CTLs that specifically recognize the target peptide [247,248]. In recent years, IVT mRNA has been further optimized with ionizable lipid nanoparticle-mediated mRNA delivery for improved viability compared to electroporation, as well as the ability to reprogram T-cells in situ using IVT mRNA carried by polymeric nanoparticles targeted to cytotoxic T-cells [240,249,250].

Clinical trials have been initiated using IVT mRNA CAR T-cells for solid tumors, including breast cancer. A clinical trial evaluating the safety and feasibility of mRNA-transfected anti-c-Met CAR T-cells found that all patients tolerated a single intra-tumoral injection of 3 × 10^7^ or 3 × 10^8^ cells. However, no measurable clinical responses were observed (NCT01837602) [47]. The transient nature of mRNA expression may necessitate multiple infusions due to reduced CAR T-cell persistence in vivo [244]. For example, an early phase I clinical trial evaluating the same mRNA-transfected anti-c-Met CAR T-cells for breast cancer and melanoma patients planned to administer up to six doses of 1 × 10^8^ modified T-cells over a short two-week period; however, this study was unfortunately terminated due to a halt in funding (NCT03060356). Safety concerns regarding the administration of multiple T-cell doses and the large amount of T-cell product required for repeated dosing raise questions regarding the feasibility of IVT mRNA-modified T-cells, and more trials are needed to fully evaluate their efficacy in various solid tumors, particularly breast cancer.

### 6.2. Allogeneic (“Off-the-Shelf”) Therapies

Allogeneic, or “off-the-shelf”, therapies are a desirable alternative to current autologous methods for breast cancer E-ACT. Not only are allogeneic therapies more cost-effective, as cell products can be banked for future use and administered upon request, but immune cells from healthy donors also have greater cellular fitness than patient immune cells that have been through multiple rounds of cytotoxic therapies [251,252]. One of the major concerns regarding allogeneic therapies is mitigating graft-versus-host disease (GvHD). GvHD occurs due to human leukocyte antigen (HLA) mismatches between the donor and the recipient [253]. The immunocompetent donor T-cells recognize the recipient as foreign, resulting in life-threatening cytotoxic activity that could seriously harm the patients [254]. Natural killer (NK) cells and gamma delta (ɣδ) T-cells are promising candidates for allogeneic therapies in breast cancer, as both can circumvent the need for HLA matching.

#### 6.2.1. Natural Killer (NK) Cells

NK cells are innate cytotoxic immune cells that recognize targets in an antigen-independent manner and play a significant role in tumor surveillance [255,256]. The fact that NK cells do not require HLA matching makes them an attractive candidate for allogeneic CAR-NK cell therapies [256]. NK cells can be isolated from three primary sources: donor peripheral blood (PB), cord blood (CB), or differentiation from CB hematopoietic stem and progenitor cells (HSPCs) or induced pluripotent stem cells (iPSCs) [252]. Although PB is an easily accessible source of NK cells, issues with donor-to-donor variability and the limited number of NK cells in a single pheresis present challenges [252,256]. CB NK cells are present at higher numbers and can be easily expanded, however, donors cannot be used again as there is a finite amount of starting material; moreover, regulations regarding the use of CB vary among countries [252,253]. Conversely, iPSCs have immense proliferative potential, are easily genetically modified, and allow for a homogenous cell product [252,253,256]. In addition to donor-derived NK cells, numerous studies have utilized the NK-92 cell line as an alternative to primary NK cells. However, NK-92 cells have reduced anti-tumor potency due to the need for irradiation [257].

Preclinical studies of CAR-NK cells for breast cancer have primarily utilized modified NK-92 cells [41,258,259,260,261,262,263] or, less frequently, PB-derived NK cells [264]. Overall, NK-92-derived and PB-derived CAR-NK cells efficiently traffic to the tumor site [41] and exhibit specific anti-tumor cytotoxicity in preclinical models [258,260,261,262,264]. Despite their potential for allogeneic therapy, both registered CAR-NK cell trials for breast cancer utilize autologous cell products (NCT05686720, NCT02839954).

#### 6.2.2. Gamma Delta (ɣδ) T-Cells

ɣδ T-cells constitute 1–5% of lymphocytes and primarily reside in epithelial tissues [251,265,266]. Most circulating ɣδ T-cells express a Vɣ9Vδ2 TCR specific for nonpeptide phosphoantigens without CD4 or CD8 coreceptors [267]. Despite their small numbers, ɣδ T-cells contribute to anti-tumor immunity through their co-expression of activating NK receptors and Toll-like receptors and their ability to lyse target cells [266]. Furthermore, ɣδ T-cells are an ideal candidate for allogeneic cell therapies because their TCR can recognize targets in an MHC-independent manner, minimizing GvHD risk [251].

Studies have demonstrated that ɣδ T-cells can target breast cancer cell lines both in vitro and in vivo [267,268,269]. We did not encounter any preclinical studies of CAR ɣδ T-cells that focus on breast cancer, however, a study using off-the-shelf anti-GPC3 CAR ɣδ T-cells controlled hepatocellular carcinoma tumor growth without evidence of GvHD [216]. One clinical trial was planned to assess allogeneic NKG2DL-targeting CAR ɣδ T-cells for various relapsed or refractory solid tumors, including TNBC, but the trial status is unknown (NCT04107142).

## 7. Current Clinical Trials

### 7.1. Trends in E-ACT Trials for Breast Cancer

A search of clinicaltrials.gov using the keywords “breast cancer”, “CAR T-cells”, and “TCR T-cells” (as well as synonyms “CAR” and “TCR”) yielded a total of 49 unique trials (as of 24 July 2023). One additional trial was identified during the literature search for a total of 50 trials. A comprehensive list of these trials is found in Appendix A. Among these trials, 32 (64%) utilized CAR T-cells, 14 (26%) used TCR T-cells, and the remaining 4 (8%) utilized engineered cells other than traditional αβ T-cells (Figure 4A). The earliest E-ACT trials involving breast cancer were initiated in 2013 (NCT01837602, NCT01967823). The University of Pennsylvania began a study to test the safety and efficacy of intratumoral injections of anti-c-Met CAR T-cells in patients with metastatic breast cancer [47]. The first TCR T-cell trial involving breast cancer targeted NY-ESO-1 and was initiated at the National Cancer Institute (NCI). During the last decade, there has been a steady increase in the cumulative number of E-ACT trials, with distinct peaks around 2015/2016 and 2020 (Figure 4B). Among the E-ACT trials to date, 29 (58%) are in Phase I, 15 (30%) are in Phase I/Phase II, and only three (6%) are in Phase II (Figure 4C). No Phase III trials have been initiated, likely due to a lack of positive results that warrant a Phase III trial. As of 24 July 2023, 23 trials are recruiting (46%), six have been completed (12%), and a total of 18 have been terminated (6, 12%), withdrawn (3, 6%), suspended (2, 4%), or their status is unknown (7, 14%) (Figure 4D). Lymphodepletion is associated with the augmented function of adoptively transferred immune cells, as it expands tumor-reactive T-cells and suppresses endogenous T_regs_ [270]. Most current E-ACT trials for breast cancer include lymphodepletion in their protocols (48%), and only two trials (4%) have chosen to omit lymphodepletion for reasons not cited (Figure 4E).

A wide variety of antigens are targeted in these trials. Unsurprisingly, the most common CAR T-cell target for breast cancer is HER2 (25%). Patients’ tumors are only considered HER2+ if they have a HER2 immunohistochemistry (IHC) score of 3+ or an IHC score of 2+ with positive fluorescence in-situ hybridization (FISH) [3]. Anti-HER2 CAR T-cells are an attractive alternative for HER2-low breast cancers, i.e., breast cancers with a HER2 IHC score of 2+ without a positive FISH or a score of 1+. Other CAR T-cell targets being investigated include MUC1 (18.8%) and mesothelin (15.6%), both of which have demonstrated promising results in preclinical studies [51,53,57,271]. Compared to CAR T-cells, TCR T-cells have the advantage of targeting intracellular CGAs and neoantigens that would otherwise be inaccessible to CAR T-cells. CGAs and neoantigens also tend to have more restricted expression, minimizing potential on-target/off-tumor toxicities [32]. The most common TCR T-cell targets include NY-ESO-1 (36%), MAGE-A3 (14%), and KK-LC-1 (14%). A comprehensive list of antigen targets for both CAR T and TCR T trials can be found in Figure 4G. Unlike CAR T-cells, successful TCR T-cell treatment requires that the patient possesses compatible HLA alleles. Among TCR-T trials, HLA-A*02 was the most prevalent restricting HLA (50%), followed by HLA-A*01 (21.4%) (Figure 4F). Most E-ACT trials for breast cancer utilize autologous cell products, apart from two current trials evaluating allogeneic CAR T-cells (NCT05239143) and CAR ɣδ T-cells (NCT04107142). These “off-the-shelf” therapies will significantly reduce patient waiting times and minimize the labor required to manufacture a single dose of CAR T-cells. Furthermore, some trials are exploring engineered cell products derived from other cell types, including CAR NK cells (NCT02839954, NCT05686720) and CAR macrophages (NCT04660929) targeting MUC1, mesothelin, and HER2, respectively.

Most breast cancer E-ACT trial locations are in the United States (58%), followed by China (36%), Malaysia (2%), Israel (2%), and Japan (2%) (Figure 4H). Among the 29 E-ACT trials held in the United States, 41.4% are sponsored by academic institutions, 34.5% by industry, and the NIH sponsors 24.1%. Most trials opened at academic institutions are from the University of Pennsylvania (3 trials, 25%), followed by Memorial Sloan Kettering Cancer Center, Fred Hutchinson Cancer Center, and Baylor College of Medicine, each with two trials. (Figure 4I). Each of the trials sponsored by the industry is supported by a different company, as listed in Figure 4I. The NCI initiated all NIH trials. In the earlier years of breast cancer E-ACT trials (2013–2017), trial sponsorship primarily came from academic institutions and the NIH. Since 2018, however, the number of industry-sponsored trials has increased, which helps to accelerate the initiation of new trials.

Trial results were located via analysis of published manuscripts. We identified seven trials that have published results from breast cancer patients. Clinical results of four CAR and three TCR T-cell trials are detailed in Table 3. We provide a broad analysis of the results from these trials, with particular emphasis on their safety and feasibility. Of note, all seven trials are in the early phase and therefore have a relatively small sample size. Despite their small numbers, the results of these early trials are a crucial step toward the clinical establishment of these products and will guide the conduct and design of future trials.

### 7.2. Safety and Efficacy of CAR T-Cells for Breast Cancer

#### 7.2.1. c-Met-Specific CAR T-Cells: A Safe and Moderately Effective Target

Two CAR T-cell trials with published results for breast cancer patients target c-Met. An initial phase 0/phase I study evaluated the safety and feasibility of intratumoral (I.T.) injections of autologous c-Met CAR T-cells in patients with metastatic breast cancer (four TNBC, two ER^+^HER2^−^). Three patients received dose level 1 (DL1) at 3 × 10^7^ cells, and three received DL2 at 3 × 10^8^ cells. It was not specified whether lymphodepletion was used. No measurable clinical responses were observed, and all grade 3 serious adverse events (SAEs) were deemed unrelated to the study drug [47]. CAR mRNA was detected in the peripheral blood or tumor tissue in 5 of 6 (83.3%) patients, however, levels became Undetectable 24 h after I.T. injection. These initial results demonstrated the safety of autologous c-Met CAR T-cells. Earlier this year, the same group published results from an additional early phase I study evaluating the safety and efficacy of intravenous (I.V.) c-Met CAR T-cells. This study included four TNBC patients and three melanoma patients. Patients received six doses of 1 × 10^8^ cells each over 14 days without lymphodepletion. Published study results observed 4 of 7 (57.1%) patients with stable disease (SD) and 3 of 7 (42.8%) with progressive disease (PD). Among the four TNBC patients, two had SD, and two showed PD. Of the seven total patients, six (85.7%) experienced grade 1/2 toxicity, and one (14.3%) experienced grade 1 cytokine release syndrome (CRS). No grade 3 or higher toxicity, neurotoxicity, or treatment discontinuation occurred [272]. Together, the results from these two trials demonstrate the safety and therapeutic potential of c-Met CAR T-cells. Additional trials and preclinical studies are warranted to fully evaluate their clinical efficacy.

#### 7.2.2. ROR1-Specific CAR T-Cells: Initial Safety and Poor Intratumoral Persistence

One phase I study evaluated the safety, persistence, trafficking, and preliminary antitumor activity of ROR1-specific CAR T-cells in patients with ROR+ solid and hematologic malignancies. A total of 21 patients with various tumor types participated in this trial and were divided among three DLs with lymphodepletion. A subsequent manuscript describes the results of three metastatic TNBC patients and will be the focus of this analysis [273]. Among these patients, 1 of 3 (33.3%) progressed after treatment, while 2 of 3 (66.7%) had SD, with one patient achieving a partial response (PR) after advancing from DL2 to DL3. Only one TNBC patient experienced grade 1 CRS. As for CAR T-cell expansion and trafficking in mTNBC patients, 2 of 3 (66.7%) experienced a robust CAR T-cell expansion in their peripheral blood, with no toxicity to normal tissues. At peak expansion, however, these T-cells upregulated inhibitory receptors and lost the ability to produce crucial effector cytokines. Unfortunately, CAR T-cells did not accumulate or persist at the tumor site. The success of ROR1-specific CAR T-cells may require additional cytokine stimulation or co-stimulatory receptors. One such trial is currently recruiting and incorporates membrane-bound IL-15, intrinsic PD-1 blockade, and a kill switch for additional safety (NCT05694364).

#### 7.2.3. Mesothelin-Specific CAR T-Cells: Emerging Results from an Ongoing Trial

A phase I/II trial evaluating regional delivery of mesothelin-specific CAR T-cells in patients with malignant pleural disease is ongoing (NCT02414269), and phase I results that include one breast cancer patient were recently published [274]. This patient received 3 × 10^5^ CAR T-cells per kilogram intrapleurally via intervention radiology-guided imaging following lymphodepletion. No adverse events were noted, and the patient received multiple lines of therapy after CAR T-cells and survived 11 months. These preliminary results suggest that mesothelin is a safe target that warrants further clinical investigation.

### 7.3. Safety and Efficacy of TCR T-Cells for Breast Cancer

#### 7.3.1. NY-ESO-1-Specific TCR T-Cells: Additional Clinical Data Needed

While NY-ESO-1 is by far the most clinically evaluated target for TCR T-cell therapy in breast cancer, few studies report results from breast cancer patients. However, one study published in 2022 provides a small window into the efficacy of NY-ESO-1 redirected T-cells in breast cancer. This phase I study tested NY-ESO-1-specific TCR T-cells in HLA-A*2:01 or HLA-A*2:06 positive patients, including one breast cancer patient treated with one dose of 5 × 10^8^ cells with lymphodepletion (NCT02366546). NY-ESO-1 expression for this patient was between 5–25%, as determined by IHC. Although the patient did not develop CRS, the transferred T-cells did not expand well in peripheral blood, and the patient progressed soon after treatment and passed away at six months post-treatment [276]. However, three other patients enrolled in the study showed greater than 30% tumor regression, albeit with much higher NY-ESO-1 expression (>/=75%). The poor response of the breast cancer patient, therefore, is likely due to low antigen density. A larger breast cancer patient population is needed to assess the safety and efficacy of NY-ESO-1-specific TCR T-cells accurately.

#### 7.3.2. MAGE-A3-Specific TCR T-Cells: Toxicity and No Evidence of Efficacy in Breast Cancer

Early reports of TCR T-cell trials targeting MAGE-A3 describe severe cardiac toxicity due to the cross-reactivity of the TCR to normal cardiac proteins [21]. A later study aimed to target MAGE-A3 in patients with metastatic cancer who were HLA-DPB1*04:01 positive (NCT02111850). The primary objective was to determine the maximum tolerated dose of TCR T-cells. Two breast cancer patients were enrolled in the high-dose cohort, receiving 7.8 × 10^10^ (plus three doses of IL-2) and 9 × 10^10^ (plus one dose of IL-2) cells with lymphodepletion. Of note, the patient treated with 9 × 10^10^ T-cells and one dose of IL-2 experienced grade 4 toxicities, including elevated ALT, AST, and creatinine. The patient later developed respiratory failure, requiring hospitalization. Both breast cancer patients exhibited no response (NR) to treatment [275]. A similar study was conducted in patients with HLA-A*01; however, it was terminated due to slow, insufficient accrual (NCT02153905). Preliminary results for only three participants are posted, and it cannot be determined if any are breast cancer patients.

#### 7.3.3. Neoantigen-Specific TCR T-Cells: Promising Results from an Ongoing Trial

An ongoing phase II clinical trial recently published results from a chemo-refractory breast cancer patient treated with T-cells transduced with an HLA-A*02:01-restricted TCR specific for the p53 R175H mutation. TCR T-cells were administered following standard lymphodepletion with no IL-2. The patient also received one dose of pembrolizumab post-ACT. After infusion, the patient developed grade 3 acute CRS, which resolved following intubation, vasopressors, and steroid treatment. Despite initial distress, the patient exhibited a 55% decrease in tumor burden at 14 weeks post-treatment. By day 60, metastatic sites had decreased, and all detectable skin lesions had resolved. Infused R175H-TCR T-cells also persisted and developed into memory T-cells that could be detected four months post-treatment. While the patient initially had a PR, they progressed at six months post-treatment. New lesions were found to have lost expression of HLA-A*02:01, thereby allowing the tumor cells to escape R175H-TCR T-cell recognition [116]. While these results warrant further investigation into preventing tumor immune escape through HLA loss, the initial anti-tumor efficacy of R175H-TCR T-cells is very promising.

## 8. Conclusions

E-ACTs for breast cancer are rapidly evolving, with no shortage of targets to explore. While great strides have been made in the preclinical assessment of these novel therapies, including technologies to improve T-cell persistence and target the immunosuppressive TME, most preclinical models are limited in their ability to recapitulate the full scope of interactions between the breast microenvironment and patient immune system. Various studies have been performed in syngeneic mouse models, utilizing murine breast cancer lines in immunocompetent mice and allowing for a comprehensive study of E-ACTs in the context of a complete TME [61,162,185,192,199,218,273,277]. However, these studies require the use of fully murine receptors that cannot be immediately translated to clinical trials without humanization. Fully humanized mice with an intact immune system can be achieved through transplantation of human CD34+ hematopoietic stem/progenitor cells [278]. While humanized breast cancer models have been developed [279,280,281], further studies are needed to establish protocols for breast cancer E-ACT in these models.

Despite the limitations of preclinical models, we present a wealth of evidence supporting the efficacy of E-ACTs for breast cancer. The real test, however, lies in their clinical efficacy. Available phase I/II trial results for patients with breast cancer demonstrate the general safety of CAR T-cell therapies; however, we eagerly await the analysis of results from numerous ongoing and completed trials. Early trials of TCR T-cell therapies for breast cancer report varied results, with some therapies showing great promise and others resulting in severe toxicities. Overall, while there are 14 CAR/TCR trials listed as completed, terminated, or suspended, results regarding breast cancer patients from only seven trials could be located. Regardless of the success of the trial, all results must be published upon trial closure. The results of these trials will guide future research and foster advancements in the technology and safety of E-ACT. Given the steady rise in the number of clinical trials using E-ACT for breast cancer since 2013, we can anticipate an influx of results in the coming years that will influence the next generation of cellular therapies. In the meantime, new studies with “armored” next-generation engineered T-cells are emerging, promising improved T-cell performance and clinical outcomes. The continued efforts to understand and overcome the factors that limit the efficacy of these therapies in solid tumors will light the way toward success.

## Figures and Tables

**Figure 1 cancers-16-00124-f001:**
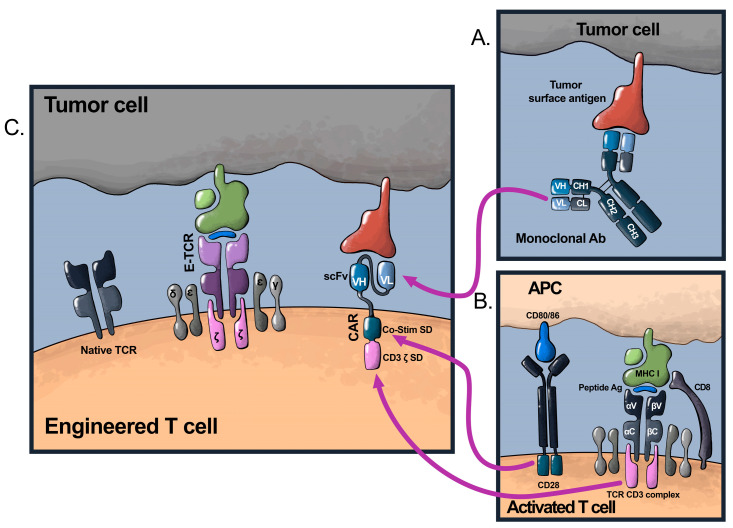
Structural components of CARs vs. E-TCRs. (**A**) Illustrates the binding of monoclonal antibodies to their respective surface antigen molecule via their variable heavy and light chains. (**B**) Illustrates the physiologic activation of non-engineered T-cells, which rely on the binding of the variable regions of their α and β chains to a peptide-MHC complex in the surface of the antigen-presenting cell and requires the other CD3ε/ɣ/δ/ζ subunits to signal. For complete activation, the T-cell also receives signaling from co-stimulatory receptors (in this case, CD28). (**C**) Shows on the right a second-generation CAR, which combines the antigen-recognition components of a mAb with the activation and co-stimulatory signals of a TCR. CARs, therefore, can recognize antigens in an MHC-independent manner. On the left, we show E-TCRs, which function in the same manner as the native TCR, requiring MHC-peptide recognition and the presence of the CD3ε/ɣ/δ/ζ subunits to signal, and lack inclusion of co-stimulation.

**Figure 2 cancers-16-00124-f002:**
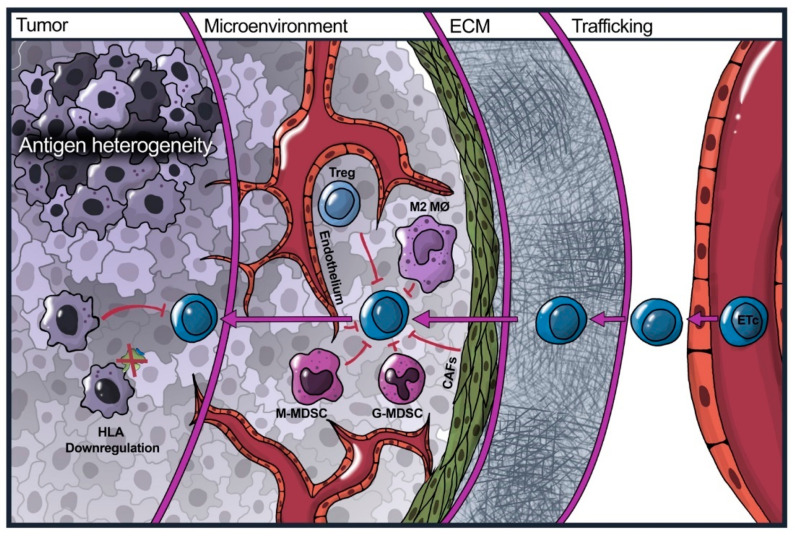
Challenges faced by E-ACTs in the treatment of solid malignancies. From right to left, E-ACTs must traffic and extravasate to the tumor site, infiltrate the ECM, and persist in the immunosuppressive tumor microenvironment before finally encountering their target. Each level presents its own set of unique challenges. All the illustrated cellular components of the TME (Stromal cells, endothelial cells, M2 macrophages, monocytic and granulocytic myeloid-derived suppressor cells, regulatory T-cells, and tumor cells) each play their role in the inhibition of T-cell effector functions.

**Figure 3 cancers-16-00124-f003:**
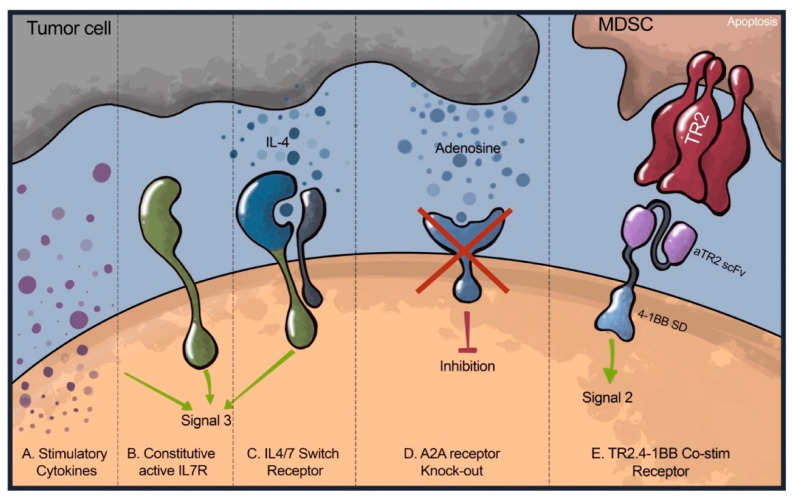
Strategies to further engineer T cells to improve their efficacy. Methods that have been explored to increase the persistence and functionality of CAR T-cells include (**A**) Incorporating a cytokine transgene allowing additional production of stimulatory cytokines (IL-15, IL-7, and IL-18), (**B**) Constitutively active IL-7 receptor which signals without the need for IL-7 cytokine binding, (**C**) engineering switch receptors that turn an inhibitory signal into a positive stimulating one, such as an IL-4 receptor with an IL-7 signaling domain, (**D**) knockout of receptors that transmit inhibitory signals to T-cells, such as the A2aR receptor, which inhibits T-cells in the presence of adenosine, and (**E**) the costimulatory TR2.4-1BB receptor that induces activation of TRAIL-R2, thereby leading to apoptosis of MDSCs while delivering a co-stimulatory signal through the 4-1BB endodomain.

**Figure 4 cancers-16-00124-f004:**
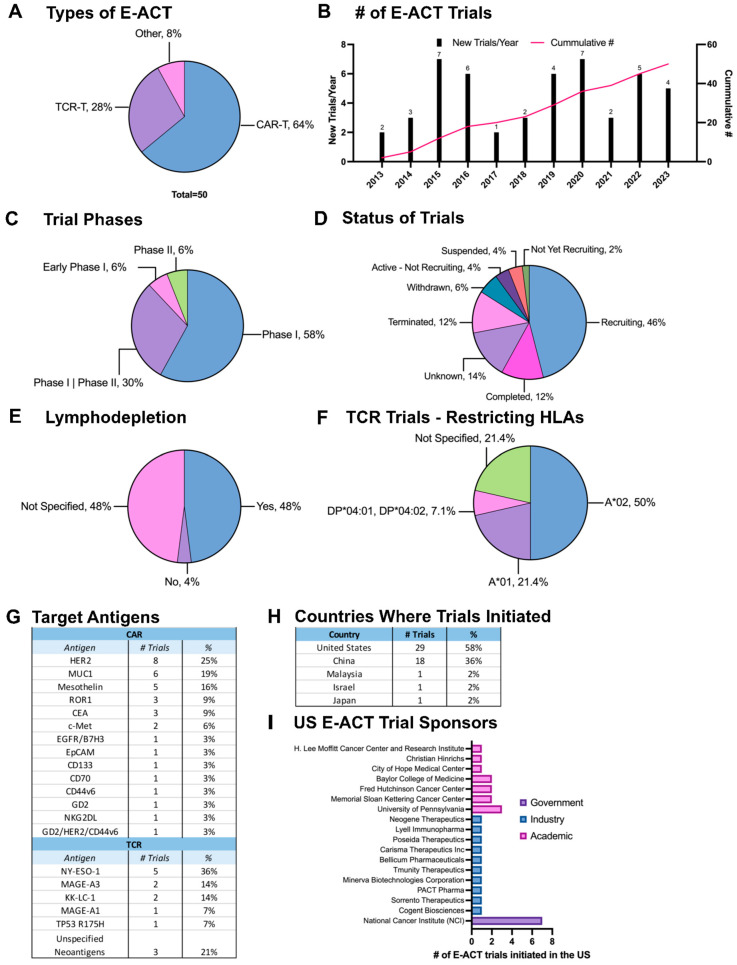
Current trends of E-ACT clinical trials for breast cancer. E-ACT trials registered in clinicaltrials.gov were assessed as of 24 July 2023. (**A**) Types of E-ACT trials. (**B**) The number of new E-ACT trials initiated each year and the cumulative number of registered E-ACT trials by year. (**C**) Phases of the 50 E-ACT trials. (**D**) Clinical status of the 50 E-ACT trials. (**E**) Use of lymphodepletion in E-ACT trials. (**F**) Restricting HLAs of the 14 TCR T trials. (**G**) Frequency of targets in E-ACT trials. (**H**) Locations where E-ACT trials have been conducted by country. (**I**) Primary sponsors of the 29 E-ACT trials conducted in the United States.

**Table 1 cancers-16-00124-t001:** Comparison of TCR and CAR T-cell constructs.

	TCR T	CAR T
Constructs	Minimally engineered TCR	Fully synthetic receptor
MHC Restriction	Dependent	Independent
Affinity and Sensitivity	Lower affinity, higher sensitivity	Higher affinity, lower sensitivity
Antigens Recognized	Peptides presented within the MHC molecule (proteins)	Cell surface proteins/molecules
Origin of Antigens	Intra-/Extracellular	Cell surface
Co-stimulatory Molecules	Endogenous CD28, 4-1BB	Linked to scFv (CD28, 4-1BB in combination with CD3ζ
Probability of CRS	Lower	Higher
References	[8,9,10]	

**Table 2 cancers-16-00124-t002:** Common breast cancer TAA, CGA, and TSA targets.

Type of Antigen	Target	Prognostic/Clinical Association	Expression in Breast Cancer	Translational Status	Ref.
TAA	HER2	Overexpression promotes tumor proliferation, migration, and survival	HER2+ (overexpression): ~20%|HER2-low: ~45–55%	Preclinical studies Clinical trials	[3,29,36,37,38,39,40,41,42,43,44]
	c-Met (HGFR)	Chemotherapy resistance|Poor survival|Increased tumor migration, invasion, and proliferation	~50% of breast cancer	Preclinical studies Clinical trials	[45,46,47,48]
	MUC1	Hypo-glycosylated in tumor cells|Associated with tumor invasion, metastasis, and angiogenesis	>90% of breast cancer	Preclinical studies Clinical trials	[49,50,51,52,53]
	Mesothelin	Metastasis|Decreased survival	67% of TNBC	Preclinical studies Clinical trials	[54,55,56,57,58,59]
	EpCAM	Worse overall survival (all cases)|Unfavorable prognosis (basal-like/luminal B HER2+)|Favorable prognosis (HER2+)	65% ER−|43% ER+|54% HER2+|47% HER2-	Preclinical studies Clinical trials	[60,61,62]
	ROR1	Aggressive disease|Tumor cell growth and survival	~40% of breast cancer 22–57% of TNBC	Preclinical studies Clinical trials	[63,64,65,66,67,68,69]
	CEA	Higher tumor burden|Poor overall survival	Elevated serum levels in 10.9–16.7% of patients	Clinical trials	[70,71]
	NKG2DL	Induced by malignant transformation of cells|May result in favorable outcomes	MIC-AB: 50%|ULBP-1: 90%|ULBP-2: 99%|ULBP-3: 100%|ULBP-4: 26%|ULBP-5: 90%	Preclinical studies Clinical trials	[72,73]
	CSPG4	Disease recurrence|Poor overall survival|Tumor migration, invasion, angiogenesis, and metastasis	77% of breast cancer	Preclinical studies	[74,75,76]
	FRα	Poor outcomes (early recurrence)	30% of breast cancer|70–80% of stage IV metastatic TNBC	Preclinical studies	[77,78,79,80]
	Ganglioside GD2	Stem cell marker|Tumorigenesis and migration	35% of breast cancer	Preclinical studies Clinical trials	[81,82,83,84]
	EGFR	Poor prognosis|Poor disease-free survival (high EGFR copy number)	61.2–64% of TNBC	Preclinical studies Clinical trials	[85,86,87,88]
	ICAM-1	Promotes bone metastasis|Aggressive phenotype|Metastasis|Poor prognosis	Overexpressed in TNBC (% not specified)	Preclinical studies	[62,89,90,91]
	CD24	Advanced stage|Shorter survival|Resistance to chemotherapy	Highest expression seen in HER2+ and TNBC samples (% not specified)	Preclinical studies	[92]
	AXL	Tolerance of chromosomal instability|Therapy resistance|Reduced survival|Supports EMT and metastasis	Overexpressed (% not specified)	Preclinical studies	[93,94,95,96,97,98]
CGA	NY-ESO-1	High humoral immune response|No association with overall survival or progression-free survival	17–28.6% TNBC|12.5% HER2+	Preclinical studies Clinical trials	[99,100,101,102]
	MAGE-A3	Worse prognosis|Reduced overall survival	~10–15% of breast cancer	Preclinical studies Clinical trials	[103,104]
	MAGE-A1	Lower overall survival	~6% of breast cancer	Clinical trials	[105]
	KK-LC-1	TNBC cell stemness|Poor survival|Malignant cell behavior	75% of TNBC	Clinical trials	[106,107,108]
TSA	PIK3CA H1047L	Cell transformation|Tumor proliferation|Resistance to apoptosis|Detected in tumors with favorable characteristics	PIK3CA mutations: 30–40% of breast cancer|~4% of PIK3CA mutations are H1047L	Preclinical studies	[34,109,110,111,112]
	TP53 R175H	Cell migration/invasion through enhanced EGFR activation|Supports tumor microenvironment|Poor survival	TP53 mutations: ~30% of breast cancer, 50% of inflammatory breast cancer|TP53 R175H: 7% of breast cancer	Preclinical studies Clinical trials	[113,114,115,116,117]

**Table 3 cancers-16-00124-t003:** Available breast cancer E-ACT trial results.

	Year	Trial ID	Target	Total # Pts	Comments	Phase	Responses	Adverse Effects	Ref.
CAR	2017	NCT01837602	c-Met	6	Intratumoral administration; mRNA electroporated CAR T-cells; Tumors resected two days later	0/I	Clinical response was not measured	All grade 3 SAEs were deemed unrelated to the study drug	[47]
	2023	NCT03060356	c-Met	7	mRNA electroporated CAR T-cells; Up to six infusions of CAR T-cells without LD; CAR T not found in tumor biopsy; 4 TNBC patients.	I	4/7 = 57.1% SD	No grade 3 or higher toxicity	[272]
	2021	NCT02706392	ROR1	21	CAR T-cells were seen in tumor biopsy; CAR T-cells upregulated inhibitory receptors and lost the ability to produce effector cytokines; 3 TNBC patients	I	2/21 = 9.5% SD	All patients noted as experiencing adverse events	[273]
	2021	NCT02414269	Mesothelin	27	LD, Intrapleural administration, +Pembrolizumab; 1 BC patient	I/II	56% SD; ORR: 12.5% (PR); BC patient did not respond	No adverse events were noted	[274]
TCR	2017	NCT02111850	MAGE-A3	17	LD and high-dose IL-2 were given; 2 BC patients	I/II	ORR: 23.5% 5.9% (CR), 17.6% (PR) BC patients did not respond	Transient G3 transaminitis (2 pts)	[275]
	2022	NCT01967823	NY-ESO-1	9	LD; 1 BC patient	I	ORR: 33% (PR) BC patient did not respond	1 Grade 3 Lung injury 3 CRS	[276]
	2022	NCT03412877	p53 R175H	1	LD; Pembrolizumab given on day 16 after TCR T-cells	II	55% decrease in tumor burden. Progressed six months post-treatment due to loss of HLA expression	Grade 3 acute CRS, resolved	[116]

LD = lymphodepletion, ORR = objective response rate, SD = stable disease, CR = complete response, SAE = serious adverse event, CRS = cytokine release syndrome.

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
