# Peer review of "Engineered Adoptive T-Cell Therapies for Breast Cancer: Current Progress, Challenges, and Potential"

_cancers, 2023, doi:10.3390/cancers16010124_

Round 1

Reviewer 1 Report

Comments and Suggestions for Authors

 Diego Chamorro et al reviewed progress and challenges Chimeric Antigen Receptor (CAR) and T-cell Receptor (TCR) T-cell therapies in targeting breast cancer. This study is interesting and give insight into adoptive T-cell therapies for breast cancer, and provide knowledge about working mechanism and clinical results of CAR-T. I have some concerns:

1.      It is unlikely that CAR T-cell treatment alone will successfully eradicate a tumor CAR T-cell therapy may be used in conjunction with small-molecule inhibitors or monoclonal antibodies, the related research should also be summarized.

2.     Limited infiltration of CAR-T cells into tumor is a major obstacle, novel advancements in promoting the trafficking and penetration of CAR-T cells, for example, enhancing the expression of some chemokine receptors were used in animal models. should be mentioned.

3.     The strengths and weaknesses of current preclinical models need to be mentioned, including human-derived CAR T cells in immunodeficient mice and humanized mouse model.

4.     Some research dissected the tumor microenvironment (TME) of breast cancer by single-cell RNA-seq, which could help understand the complexity and interaction of TME, which could also be referred to in the review.

Reviewer 2 Report

Comments and Suggestions for Authors

This review is well-written and contains a wealth of information regarding the basic, translational and clinical studies involving adoptive T cell therapy against breast cancer. The authors have a comprehensive summarization of the studies in recent years and provide a balanced view on the progresses being made and challenges to be resolved. Besides a few minor typo/grammatic errors, the manuscript meets the criteria of acceptance for publication in Cancers. 

Author Response

We would like to thank the reviewers and editors for their thoughtful and detailed review of our manuscript. As per the editor’s request we are submitting a revised manuscript with the tracked changes and this letter replying to each point raised by the reviewers. With the resubmission, we believe that we have comprehensively addressed all your comments.

This review is well-written and contains a wealth of information regarding the basic, translational and clinical studies involving adoptive T cell therapy against breast cancer. The authors have a comprehensive summarization of the studies in recent years and provide a balanced view on the progresses being made and challenges to be resolved. Besides a few minor typo/grammatic errors, the manuscript meets the criteria of acceptance for publication in Cancers. 

We thank the reviewer for their kind and thoughtful comments. Prior to submission of our revised manuscript, we have proofread the document for and correct any grammatical errors, per the reviewer’s request.